# All-Trans Retinoic Acid Attenuates Fibrotic Processes by Downregulating TGF-β1/Smad3 in Early Diabetic Nephropathy

**DOI:** 10.3390/biom9100525

**Published:** 2019-09-25

**Authors:** Edith Sierra-Mondragon, Rafael Rodríguez-Muñoz, Carmen Namorado-Tonix, Eduardo Molina-Jijon, Daniel Romero-Trejo, Jose Pedraza-Chaverri, Jose L. Reyes

**Affiliations:** 1Department of Physiology, Biophysics, and Neurosciences, Center for Research and Advanced Studies of the National Polytechnic Institute (Cinvestav-IPN), Laboratory 36, México DF 07360, Mexico; bio_edithsierra@hotmail.com (E.S.-M.); rafaelr@fisio.cinvestav.mx (R.R.-M.); cnamora@fisio.cinvestav.mx (C.N.-T.); 2Glomerular Disease Therapeutic Laboratory, Department of Internal Medicine, Rush University Medical Center, Chicago, IL 60612, USA; Eduardo_MolinaJijon@rush.edu; 3Department of Physiology, Biophysics, and Neurosciences, Center for Research and Advanced Studies of the National Polytechnic Institute (Cinvestav-IPN), Laboratory 44, México DF 07360, Mexico; dromeroipn@hotmail.com; 4Department of Biology, Faculty of Chemistry, National Autonomous University of México (UNAM), University City D.F. 04510, Mexico; pedraza@unam.mx

**Keywords:** extracellular matrix, renal fibrosis, retinoic acid receptor, glomeruli, proximal tubules

## Abstract

Diabetic nephropathy (DN) involves damage associated to hyperglycemia and oxidative stress. Renal fibrosis is a major pathologic feature of DN. The aim of this study was to evaluate anti-fibrogenic and renoprotective effects of all-trans retinoic acid (ATRA) in isolated glomeruli and proximal tubules of diabetic rats. Diabetes was induced by single injection of streptozotocin (STZ, 60 mg/Kg). ATRA (1 mg/Kg) was administered daily by gavage, from days 3–21 after STZ injection. ATRA attenuated kidney injury through the reduction of proteinuria, renal hypertrophy, increase in natriuresis, as well as early markers of damage such as β2-microglobulin, kidney injury molecule-1 (KIM-1), and neutrophil gelatinase-associated lipocalin (NGAL). The following parameters increased: macrophage infiltration, localization of alpha-smooth muscle actin (αSMA)-positive cells in renal tissue, and pro-fibrotic proteins such as transforming growth factor-β (TGF-β1), laminin beta 1 (LAM-β1), and collagens IV and I. Remarkably, ATRA treatment ameliorated these alterations and attenuated expression and nuclear translocation of Smad3, with increment of glomerular and tubular Smad7. The diabetic condition decreased expression of retinoic acid receptor alpha (RAR-α) through phosphorylation in serine residues mediated by the activation of c-Jun N-terminal kinase (JNK). ATRA administration restored the expression of RAR-α and inhibited direct interactions of JNK/RAR-α. ATRA prevented fibrogenesis through down-regulation of TGF-β1/Smad3 signaling.

## 1. Introduction

Diabetic nephropathy (DN) is considered a long-term diabetes mellitus complication with proteinuria, a progressive decline in renal function accompanied by glomerular and interstitial fibrosis. It is considered to be the leading cause of dysfunction in end-stage renal disease (ESRD) [1,2]. The fibrotic process is characterized by sustained inflammation, including inflammatory cell infiltration and secretion of cytokines, accumulation and imbalance of extracellular matrix (ECM), with degradation and activation of myofibroblasts [3,4,5]. Activation of transforming growth factor-1 (TGF-β1) signaling pathway has been demonstrated to play a detrimental role in the pathogenesis of progressive renal fibrosis [6]. Active TGF-β1 associates with its cell membrane type I and type II serine/threonine kinase receptors (TβRI and TβRII) with downstream phosphorylation in Smad 3 and Smad 2 (Mothers against DPP homology proteins). Phosphorylated Smad3 forms a complex with Smad4 that translocates into the nucleus to regulate the expression of numerous genes that promote fibrosis [6,7,8]. Smad7 is an inhibitor of Smad proteins that blocks the phosphorylation of Smad2 and Smad3, promoting its degradation [6,9,10]. Therapeutic approaches based on inhibiting TGF-β/Smad signaling have been reported to reduce renal injury and fibrosis in several pathological models, whereas overexpressing TGF- β1 induces renal fibrosis [6,8,11]. Renal fibrosis has been associated with the development of ESRD, therefore, advances in our understanding of renal fibrogenesis might lead to development of earlier therapeutic interventions in DN.

All-trans retinoic acid (ATRA), an active metabolite of vitamin A, belongs to the retinoids family. ATRA plays a key role in nephrogenesis. It is essential in branching of the ureteric bud and contributes to the final nephron number [12]. The pleiotropic activities of ATRA are mediated by two classes of nuclear receptors, retinoic acid receptor (RAR; α, β and γ) and retinoid X receptors (RXR; α, β and γ), leading to activation of retinoic acid response elements (RARE) on target genes [13]. ATRA has a beneficial role on the progression of renal diseases. Its use is often associated with the reduction in proteinuria and improved glomerular and tubular lesions [14,15]. We previously reported that ATRA is a nephroprotective agent in a model of acute renal failure [16], and its administration exerts antioxidant effects that prevented the loss of renal tight junction proteins, claudins 5 and 2 in DN [17]. We also demonstrated the anti-inflammatory effect ATRA by suppressing the toll-like receptor 4 (TLR4) and the nuclear factor-кB (NF-кB) signaling in glomeruli and proximal tubules in a model of diabetes [18,19]. It has been reported that some ATRA signaling components are involved in the evolution of diabetic pathology. Basu and Basualdo et al. [20] reported low plasma concentrations of vitamin A in diabetic patients, and in agreement with this finding, they described low quantities of retinol binding protein (RBP) in renal tissue of diabetic rats. Starkey et al. [21] reported low concentration of retinoid acid, associated to low activity of aldehyde dehydrogenase 1 in diabetic mice. This enzyme transform retinol to retinoic acid. Guleria et al. [22] reported that RAR-α and RXR-α are downregulated in cardiomyocytes exposed to high glucose concentrations. This decreased expression of RR-α and RXR-α contributes to hyperglycemia-induced cardiac remodeling. Whether ATRA and RAR-α mediated signaling participates in the development of initial diabetic renal alterations is not fully established. 

In this study, we evaluated whether ATRA prevented early diabetic complications through blockade of fibrogenic processes in different nephron segments—glomeruli and proximal tubules. Our findings demonstrate that TGF-β/Smad3 signaling overexpression and renal fibrogenesis are not exclusive to the late stages of ESRD but are present in early stages of diabetes. ATRA ameliorated renal injury and preserved renal function during initial DN through the activation of RAR-α. We also studied the association between RAR signaling and c-jun N-terminal kinase (JNK) pathway because they have been described as contributing to hyperglycemia-induced renal fibrogenesis.

## 2. Materials and Methods 

### 2.1. Reagents

All-trans retinoic acid, streptozotocin, Percoll, bovine serum albumin (BSA), peanut oil, methanol, citrate, ammonium sulfate, Type II collagenase, phenylmethylsulfonyl fluoride (PMSF), and antibody mouse anti-TGF-β1 were purchased from Sigma-Aldrich (St. Louis, MO, USA). Primary antibodies for mouse anti-β-2-microglobulin, rabbit anti- NGAL, mouse anti-glyceraldehyde 3 phosphate dehydrogenase (GAPDH), goat anti-sodium glucose cotransporter-2 (SGLT-2), mouse anti claudin-8 (Cldn-8), rabbit anti-laminin beta 1 (LAM-β1), mouse anti-RAR-α, rabbit anti-collagen type I (COL I), mouse anti-collagen type IV (COL IV), mouse anti-monocyte chemoattractant protein-1 (MCP-1), and goat anti-kidney injury molecule-1 (KIM-1) were available from Santa Cruz Biotechnology, Inc. (Dallas, TX, USA). Rabbit anti-F4/80, rabbit anti-Smad3 (phospho serine 423/425) and mouse anti-Smad7 were obtained from Abcam (Cambridge, UK). Mouse anti-α-smooth muscle actin (α-SMA) and rabbit anti-phospho serine were available from Merck Millipore (Darmstadt, Germany). Rabbit anti-JNK (phospho threonine 183 and tyrosine 185) were from Cell Signaling Technology (Danvers, MA, USA), and rabbit anti-occludin, recombinant protein G agarose were purchased from Invitrogen (Carlsbad, CA, USA). Bio-Rad Protein Assay Kit was from Bio-Rad Laboratories (Hercules, CA, USA). IgG-free albumin (1331-A) was from Research Organics (Cleveland, OH, USA). 

### 2.2. Animals 

Female Wistar rats, weighing 200–250g, were purchased from the Experimentation Unit of Laboratory Animals (UPEAL) of CINVESTAV-IPN. The experiments were approved by UPEAL guidelines, protocol #0178-16, and in agreement to the Mexican Official Norm NOM-062-ZOO-1999. Animals were housed in a barrier environment at constant room temperature 22 °C with 12:12 dark/light cycles. Rats were fed with a standard rodent diet (PicoLa^®^Rodent Diet 20, St Louis, MO, USA), which contains 15 International Units of Vitamin A/g, and water ad libitum. 

### 2.3. Experimental Diabetic Nephropathy

Diabetes was induced by a single intraperitoneal injection of STZ at a dosage of 60 mg/kg dissolved in freshly prepared citrate buffer, pH 6.0. Non-diabetic rats were injected with an equal volume of citrate buffer. Induction of diabetes was confirmed three days after STZ injection by measuring blood glucose concentration with OneTouch Ultra blood glucose meter (Milpitas, CA, USA). Only the rats with blood glucose levels ≥ 350 mg/dl, under fasting conditions were included in the study. Blood glucose and body weight of all animal were measured on days 3, 7, 14, and 21 of study.

### 2.4. Experimental Design 

After the induction of diabetes, rats were randomly assigned into four experimental groups (n = 6–8 per group): (a) Control group (CTL), treated daily by oral route with peanut oil from days 3–21 after injection of citrate buffer; (b) diabetic group (DBT), treated with a single intraperitoneal injection of STZ; (c) diabetic group treated with ATRA (DBT+ATRA), given daily by oral route (ATRA 1 mg/Kg) from days 3–21 after STZ injection; and (d) animals control treated with ATRA, the ATRA treatment was given daily by oral route from days 3 to 21 after single injection of citrate buffer. At the end of three weeks, rats in different groups were sacrificed to obtain kidneys and blood samples. ATRA suspension was dissolved in peanut oil solution. 

### 2.5. Measurement of Renal Function and Biochemical Markers

After maintenance for three weeks, rats were housed in metabolic cages for 24 h before sacrifice to collect urine for subsequent measurements of urine protein concentration using the Lowry colorimetric method. Body and kidney weights were measured at the end of the study, and kidney hypertrophy was estimated by determination of kidney/body weight ratio. Blood samples were obtained from cardiac puncture under anesthesia, and serum was separated by centrifugation. Urinary and serum creatinine levels were measured by the modified Jaffe colorimetric method as previously described [23]. The urinary protein to creatinine ratio was obtained by dividing the urine protein concentration by the urine creatinine concentration. Tubular function, was determined by the fractional excretion of sodium (FeNa). FeNa was calculated through the following formula: FeNa = (sodium clearance/creatinine clearance) X 100 and expressed as a percent. Sodium concentrations in the serum and urine were measured by atomic absorption spectrophotometry (Perkin Elmer 3100 with an air-acetylene flame, Norwalk, CT, USA) as previously described [23,24]. 

### 2.6. Histopathology of the Kidney

Renal tissues were fixed in 10% formaldehyde, and paraffin-embedded tissue sections (4 μm) were cut and stained with hematoxylin and eosin (H&E). The glomerular volume was determined by light microscopy (Olympus BX43F microscope, Tokyo, Japan) of H&E sections. The surface area (µm2) of a minimum of 12 cortical glomeruli sections from each experimental group was determined in digital images using Image J 4.5 software. Glomerular volumes were calculated using the Weibel-Gomez formula under the correction factors as described Lane et al. [25] according to the following equation: Glomerular volume = Area1.5 × 0.75 + 0.21.

### 2.7. Isolation of Glomeruli

Renal cortices were cut into small pieces, and glomeruli were isolated by the mechanical graded sieving technique as previously described [17,18]. The homogenized tissue was pushed through a 117 µm stainless steel sieve (cat. No. 8321A44; Thomas Scientific, Swedesboro, NJ, USA). The material that remained on top of the sieve (containing an enriched fraction of glomeruli), was collected in ice cold Krebs bicarbonate solution (KB): 110 mM NaCl, 25 mM NaHCO_3_, 3 mM KCl, 1.2 mM CaCl_2_, 0.7 mM MgSO_4_, 2 mM KH_2_PO_4_, 10 mM sodium acetate, 5.5 mM glucose, 5 mM alanine, and 0.5 g/L bovine serum albumin (BSA), pH 7.4, osmolarity 290 mOsm/kg H_2_O, and centrifuged at 20,000× *g* for 10 min. The sieved tissue containing a preparation enriched in glomeruli was transferred to a second sieve with a pore opening of 74 μm (cat. no. 8321A58; Thomas Scientific). After several washings with ice-cold phosphate-buffered saline (PBS), the material that remained on top of the sieve, which contained the glomeruli, was collected in ice-cold PBS and centrifuged at 20,000× *g* for 10 min. The supernatant was decanted and washed one final time using the same centrifugation, and the pellet was resuspended in PBS with protease inhibitor cocktail (complete 1X (Boehringer Mannheim, Germany) and PMSF (1 mM)). The protein content in lysates of glomeruli was determined by the Lowry colorimetric method. Isolation of glomeruli was confirmed by Western blot analysis of the specific protein marker nephrin (Appendix A).

### 2.8. Isolation of Proximal Tubules

Renal tubules were isolated from renal cortex by Percoll density gradient centrifugation as previously described [18,24]. The cortical fragments were digested with type II collagenase (15 mg) and 0.5 mL of 10% BSA. Samples were gassed with 95% air 5% CO_2_ in a shaking water bath for 20 min at 37 °C. After digestion, 20 mL of ice-cold KB solution with a protease inhibitor cocktail were added, and the suspension was gently agitated to disperse tissue fragments. Suspension was filtered to remove collagen fibers. The pellet was resuspended in 10 mL of ice-cold KB with the protease inhibitor cocktail. This washing procedure was repeated thrice. The tissue pellet was then resuspended in 5% BSA with protease inhibitors for 5 min at 4 °C and centrifuged for 1 min, and supernatant was discarded. Tissue pellets were suspended in 30 mL of a freshly prepared mixture of ice-cold Percoll and KB (1:1, *v/v*). Thereafter, the suspension was centrifuged (1071× *g* for 30 min), resulting in the separation of three bands. The first band was enriched with distal tubules, the second band with glomeruli, and the third band contained an enriched suspension of proximal tubules. For total protein extraction, proximal tubules and distal tubules suspensions were gently centrifuged (18× *g* for 30 s). Pellets were suspended in 10 mL of ice-cold KB containing protease inhibitor cocktail. This washing procedure was repeated thrice. Tissue pellets were suspended and incubated for 30 min at 4 °C in 300 μL of lysis buffer (RIPA): 40 mM Tris-HCl pH 7.6, 150 mM NaCl, 2 mM EDTA, 10% glycerol, 1% Triton X-100, 0.5% sodium deoxycholate, 0.2% sodium dodecyl sulfate (SDS), 1 mM sodium orthovanadate, 0.5 mM sodium fluoride, 1 mM PMSF, and complete 1X. Thereafter, samples were sonicated thrice for 30 s each in a high-intensity ultrasonic processor (Vibra cell; Sonics & Materials Inc., Danbury, CT, USA), centrifuged at 20,000× *g* at 4 °C for 40 min, and supernatants were collected. Isolated tubules were confirmed by Western blot analysis of specific protein markers: SGLT2 for proximal tubule and claudin-8 for distal tubule (Appendix A).

### 2.9. Western Blot Analysis

Protein samples (60–80 μg) or a volume of urine from each animal corresponding to the same excretion fraction up to 24 µL of urine were separated by sodium dodecyl sulfate (SDS)-polyacrylamide gel electrophoresis and transferred to polyvinylidene difluoride (PVDF) membranes (Millipore Corp. Bedford, MA, USA). Membranes were then blocked in phospathe-buffered saline containing 3% BSA and incubated with primary antibodies, including anti-β2 microglobulin (1:500), MCP1 (1:1000), α-SMA (1:1000), LAM β1 (1:500), COL IV (1:2000), COL I (1:500), TGF-β1 (1:500), phospho Smad3 (1:2000), RAR-α (1:2000), phospho serine (1:1000), Smad 7 (1:2000), GAPDH (1:1000), and phospho JNK (1:2500). The proteins were detected using respective peroxidase-conjugated secondary antibodies (1:10,000) at room temperature, and chemiluminescences were developed (UVP Biomaging Systems, Cambridge, UK). Quantification was performed by measurement of signal intensity with Image J software (National Institute of Health, Bethesda, MD, USA).

### 2.10. Immunohistochemistry Analyses

Renal tissue sections were used to perform immunofluorescence analysis for NGAL, F4/80, α-SMA and phospho Smad3. Then, 6 µm sections were cut in a Leica CM 1510 cryostat (Wetzlar, Germany) and mounted on gelatin coated slides. The sections were fixed for 10 min with methanol and subsequently incubated for 5 min at room temperature in 1% (*v/v*) Triton X-100, then blocked with 0.5% (*w*/*v*) IgG-free albumin for 1 h at room temperature, followed by incubation overnight at 4 °C with primary antibodies anti-NGAL, anti-F4/80, anti-α-SMA, and anti-phospho Smad3 (dilution 1:100). Proximal tubules were double labeled with SGLT2 (dilution 1:300) to differentiate them from other structures. Secondary antibodies Alexa Fluor 488 donkey anti-rabbit, Alexa Fluor 488 donkey anti-goat, and Alexa Fluor 594 donkey anti-mouse (dilution 1:300) were used. Images were acquired using a confocal inverted microscope (TCS-SP2, Leica, Heidelberg, Germany). Immunofluorescence experiments were performed at least three times in samples from three animals per group. Immunolabelling was quantified as stained area (µm2) or nuclei-positively stained cells of the total cells. Quantification was estimated on captured high-quality images using Leica software TCS-SP2 (Heidelberg, Germany). Phospho Smad3 nuclear translocation was quantified as the percentage of the p-Smad3 nuclei-positively stained cells of the total cells that were counted from three randomly chosen fields from three different animals per group. From 70 to 120 nuclei were counted for each experimental condition.

### 2.11. Immunoprecipitation (IP)

IP of RAR-α was performed in isolated glomeruli and isolated proximal tubules with anti-RAR-α (10 μg of mouse anti-RAR-α). Lysates of glomeruli and proximal tubules (1 mg) were incubated overnight at 4 °C with 20 μL of recombinant protein G-agarose beads. G-agarose were removed by centrifugation at 16,000× *g* for 5 min and incubated overnight at 4 °C with 2.5 mg of the immunoprecipitating antibody previously bound to protein G-agarose. In parallel incubations, we used an irrelevant antibody (anti-occludin) as a negative control. IP proteins were then analyzed by Western blot as described above.

### 2.12. Data Analysis

Results are expressed as mean ± SD. Statistical differences between groups were analyzed by One-way analysis of variance (ANOVA) or Two-way ANOVA-ordinary. Tukey was used as a post-hoc test. The data were analyzed using the software GraphPad Prism 6.0 (GraphPad Software, Inc., La Jolla, CA, USA). *p* < 0.05 was considered statistically significant.

## 3. Results

### 3.1. ATRA Preserves Renal Function of Rats with Early DN

We studied the effect of ATRA (1 mg/kg body) on the development of DN. Four groups of rats were included in this experiment: CTL, DBT, DBT+ATRA, and ATRA. Notably, rats showed hyperglycemic levels 3 days after STZ injection in DBT (428.3 ± 37.0 mg/dl) and DBT+ATRA (421.5 ± 34.5) groups as compared to CTL (94.5 ± 4.3) and ATRA (92.1 ± 3.3) groups. Similarly, blood glucose levels were maintained in the four experimental groups until the end of the experimental period (DBT: 413.8 ± 51.6, DBT+ATRA: 404.3 ± 37.6, CTL: 89.1 ± 2.8 and ATRA: 88.3 ± 7.1), while ATRA treatment did not change the elevated glucose concentration of diabetic rats (Figure 1a). DBT group developed weight loss (Figure 1b) and renal hypertrophy (Figure 1c) as revealed by an increase in kidney weight/body weight ratio compared with nondiabetic groups. Induction of diabetes also caused an elevation of proteinuria/creatininuria ratio (Figure 1d) and in the fractional sodium excretion (FeNa, Figure 1e) compared to CTL group, whereas ATRA administration to diabetic rats remarkably improved these parameters and partially prevented the body weight loss on day 21 of diabetes (Figure 1b). These results indicate that ATRA treatment effectively ameliorate glomerular and tubular dysfunction, and its beneficial effect was independent of the glycemic control.

### 3.2. The effect of ATRA on Pathological Lesion of Early DN

In addition to renal functional results, morphological changes were identified by H&E staining in kidney samples from the four experimental groups (Figure 2a). Glomerular volume was used to assess glomerular hypertrophy. In diabetic rats the glomerular volume was increased (Figure 2b, black arrows) and histological examination revealed tubular lumen dilation (black asterisks). Moreover, ATRA treatment improved renal histopathologic changes in diabetic rats (Figure 2a,b). Early kidney injury of diabetic rats was confirmed by the measurement of biomarkers such as urinary β-2-microglobulin, KIM-1, and NGAL expression. These markers have been identified as sensitive indicators of early renal tubular injury. We found that urinary β-2-microglobulin excretion increased in diabetic rats compared to nondiabetic groups (Figure 2c). KIM-1 expression was evaluated by Western blot in isolated proximal tubules. As shown in Figure 2d, significant increases in the expression of KIM-1 was found in DBT group compared to CTL group. Immunofluorescence analysis indicated an increased expression of NGAL in proximal tubules of DBT group compared to control rats (Figure 2d,e), which was demonstrated for the co-localization of NGAL (green label) with SGLT-2 (a marker of proximal tubules brush border, red label). By contrast, ATRA treatment remarkably ameliorated these alterations. These results clearly suggest that, after three weeks of diabetes induction, there are early renal structural changes, including glomerular and tubular hypertrophy, and ATRA treatment showed nephroprotective effects in the preservation of renal morphology and in the reduction of the glomerular and tubular damage.

### 3.3. ATRA Treatment Prevents Renal Accumulation of Macrophages into Glomerular and Tubule-Interstitial Areas during Initiation of DN

Macrophages have been shown to be actively involved in fibrogenesis by providing cytokines and growth factors that modulate the proliferation and collagen synthesis of fibroblasts [26,27]. Herein, we aimed to explore the effects of ATRA on macrophage infiltration and chemoattractant factor expression. As shown in Figure 3, by confocal microscopy we found that the number of macrophages (F4/80 positively stained cells, green label) was significantly increased into glomerular (Figure 3a,b) and tubule-interstitial (Figure 3c,d) regions in diabetic group compared to nondiabetic groups, whereas ATRA treatment markedly diminished F4/80-positive macrophages infiltration in diabetic rats. Chemokines are important mediators by recruiting and activating specific immune cells, the monocyte chemoattractant protein-1 (MCP-1) is considered as a central molecule in macrophage influx in several types of kidney diseases including DN [28]. MCP-1 was assessed by Western blot in enriched nephron sections: glomeruli and proximal tubules; these segments were validated using specific markers of glomeruli and proximal tubules (Appendix A). The expression of MCP-1 was significantly increased in isolated glomeruli and proximal tubules from diabetic rats (Figure 3e,f). ATRA treatment decreased the overexpression of MCP-1 in each nephron section of DBT+ATRA group. These data suggest that ATRA inhibits early diabetes-induced glomerular and tubular injury through an anti-fibrogenic effect by reducing MCP-1 expression and macrophage infiltration.

### 3.4. ATRA Treatment Decreases Profibrotic Processes during Early DN

Myofibroblasts are a subset of activated fibroblasts, characterized by expression of α-SMA, which are the principal cell type responsible for excessive ECM deposition during renal fibrogenesis [4]. To further explore the anti-fibrogenic effect of ATRA, we analyzed the expression of α-SMA by immunofluorescence in renal tissue. Diabetes increased the localization of α-SMA-positive cells (red label) into glomeruli (Figure 4a,b) and tubulointerstitial space (Figure 4c,d) compared to CTL rats. Additionally, α-SMA expression was evaluated by Western blot in isolated glomeruli and isolated proximal tubules. α-SMA expression was enhanced in diabetic condition (Figure 4e,f). Also, we observed that ATRA treatment decreased the localization and the expression of α-SMA in glomeruli and proximal tubules in diabetic rats. These data suggest that ATRA exerts an inhibitory effect on transformation of fibroblasts into myofibroblasts during fibrogenesis process in early DN.

### 3.5. ATRA Treatment Prevents Diabetic-Induced Changes associated to the Expression of Markers of ECM Accumulation

Renal fibrosis, characterized by excessive accumulation of ECM, plays a central role in development of DN. We next investigated whether markers of ECM increased at three weeks of DN and determined the effects of ATRA on fibrogenesis in diabetic rats. Therefore, the expression of LAM β1, COL IV, and COL I in isolated glomeruli (Figure 5a) and isolated proximal tubules (Figure 5b) was examined by Western blot. Expression of these markers were increased in glomeruli and proximal tubules from DBT group, while administration of ATRA decreased the high expression of LAM β1, COL IV, and COL I induced by diabetes in glomeruli and proximal tubules. In brief, our data indicated that renal fibrosis in glomeruli and proximal tubules is induced during early DN and ATRA treatment prevents glomerular and tubular damage by decreasing these fibrogenic changes.

### 3.6. ATRA Treatment Ameliorates Fibrogenesis by Suppressing TGF-β1/Smad3 Signaling Pathway in Glomeruli from Diabetic Rats

To elucidate the mechanism of the anti-fibrotic potential of ATRA, the TGF-β1/Smad3 signaling pathway was investigated. TGF-β1 is well identified as a central mediator in renal fibrosis. TGF-β1 initiates intracellular signals through phosphorylation and nuclear translocation of Smad3. Smad3 promotes renal fibrosis by directly binding to the promoter region of ECM molecules to trigger their production [7]. In addition, TGF-β1 also induces an I-SMAD called Smad7, which negatively regulates activation of Smad3 [10]. Western blot analysis (Figure 6a) demonstrated that TGF-β1 expression in isolated glomeruli of diabetic group was increased, followed by overexpression of phospho (p)-Smad3. Whereas Smad7 expression in diabetic glomeruli did not significantly increased compared to nondiabetic groups (Figure 6a). Further immunofluorescence experiments showed nuclear translocation of p-Smad3 (red label) in glomerular cells of diabetic rats (Figure 6b,c). Interestingly, ATRA treatment enhanced Smad7 expression and decreased the expression of TGF-β1 (Figure 6a) and p-Smad3 and inhibited the nuclear translocation of p-Smad3 in glomeruli of diabetic rats (Figure 6b,c). These results indicated that the anti-fibrotic effects of ATRA were possibly related to suppress TGF-β1/Smad3 signaling pathway in glomeruli under diabetic conditions by overexpression of Smad7.

### 3.7. ATRA Treatment Attenuates Diabetes-induced Activity of TGF-β1/Smad3 Signaling in Proximal Tubule

Next, we sought to determine the role of ATRA in the regulation of TGF-β1/Smad3 signaling in isolated proximal tubules. We examined levels of TGF-β1, p-Smad3, and Smad7 by Western blot (Figure 7a). In early diabetes increased the expression of TGF-β1 and p-Smad3 were observed, while Smad7 did not show a significant increase compared to nondiabetic groups. Moreover, by confocal microscopy (Figure 7b,c) was observed that early diabetes induced nuclear translocation of p-Smad3 (red label) in proximal tubular cells compared to CTL group. SGLT2, as specific marker of proximal tubule cells (green label), was used as its indicator. In addition, we observed that ATRA decreased the level of TGF-β1 and the phosphorylation of Smad3 in serine 423/425 (Figure 7a), thus preventing its nuclear translocation in proximal cells (Figure 7b), while ATRA increased Smad7 expression (Figure 7a). The above described findings suggest that ATRA could modulate fibrogenesis process by suppression of TGF-β1/Smad3 signaling pathway in proximal tubular cells in diabetic rats.

### 3.8. ATRA Inhibits Diabetes-Induced Phosphorylation and Loss of RAR-α Expression by JNK.

The cellular action of ATRA is mediated through RAR-α, β and γ, these receptors directly activate gene transcription [12]. We previously demonstrated that ATRA is a potent nephroprotective agent; its administration exerts antioxidant effects [17] and anti-inflammatory effects during initiation of DN [18,19]. These studies suggested that changes in ATRA signaling via the extracellular/intracellular ATRA level or the expression/activation of RAR-α, closely correlate with the development of diabetes. Thus, we determined whether early diabetes affects RAR-α expression in isolated glomeruli and proximal tubules (Figure 8a,b). It was found that RAR-α expression significantly decreased in diabetic condition, which was prevented and increased by ATRA treatment (DBT + ATRA group). It has been reported that phosphorylation of RAR-α at specific serine sites leads to its degradation. JNK promotes phosphorylation of RAR-α, which further leads to proteasomal degradation and transcriptional inhibition of RAR-α [29,30]. Based on the above-described evidence, we confirm the direct interaction of these proteins; IP assays were performed in isolated glomeruli and proximal tubules (Figure 8c,d). The serine phosphorylation of RAR-α was observed at three weeks of diabetes compared to nondiabetic groups. These phosphorylations are consistent with decreased expression of RAR-α. It was found that diabetes increased immuneprecipitation of RAR-α with phospho p-JNK. These results demonstrated that diabetes induces interaction of RAR-α/p-JNK. All these changes induced during initial diabetes were decreased by ATRA treatment.

## 4. Discussion

Kidney fibrosis is a common endpoint of various progressive kidney diseases such as DN, which leads to loss of nephrons and is characterized by the activation of fibroblasts, epithelial-to-mesenchymal transition, macrophage infiltration, and excessive ECM accumulation that may lead to the development of ESRD [5,6]. Inflammation plays a critical role in the initiation and progression of renal fibrosis [1,2,3]. We have recently demonstrated that an inflammatory process is induced in initial stages of DN with increased cytokines, chemokines, cell adhesion molecules, and growth factors in glomeruli and proximal tubules [18,19]. However, further understanding of the mechanisms behind renal fibrogenesis is essential to develop therapies to prevent or reverse this process and slow its progression in DN to end-stage renal disease.

Previous studies have suggested that ATRA might protect against the development of renal pathological changes including DN [13,14,15,16,17]. Nevertheless, whether ATRA might attenuate renal fibrogenesis has not been studied in DN. Therefore, in the present study, we examined renal fibrotic events and the therapeutic effects of ATRA during early stage of diabetes in select nephron segments (glomeruli and proximal tubules).

In this study, three weeks after STZ-induction, these animals displayed symptoms of diabetes such as hyperglycemia, body weight loss, and abnormal renal function, including an increase of proteinuria/creatinuria ratio. Proteinuria is a strong functional parameter of damage in the early stage of DN and is considered to have both hemodynamic (glomerular capillary hypertension and hyperfiltration) and structural bases—changes in glomerular basement membrane, mesangial cell matrix, and podocyte function [31]. Consistent with these data, we found glomerular hypertrophy and an increase of kidney weight/body weight ratio and glomerular volume. Also, it was found tubular damage associated with increased FeNa and expression of urinary β2-microglobulin, KIM-1, and NGAL in tubular tissue. β2-microglobulin, KIM-1, and NGAL have been used as early markers for the evaluation of renal damage in patients with diabetes type 1 and type 2 as well as for the early diagnosis of diabetic nephropathy [32]. Removal of β2-microglobulin from the serum is primarily through glomerular filtration but more than 90% of the filtered protein is reabsorbed and catabolized in the proximal convoluted tubule resulting in minimal urinary concentration of β2-microglobulin under normal conditions [33]. NGAL is a 25-kDa molecule known to be hyper-produced in kidney tubules short after kidney injury initiation [34]. As expected, KIM 1 and NGAL were present in our diabetic condition, indicating the successful establishment of an early-stage DN model.

Daily oral administration of ATRA (1 mg/kg b.w.) from days 3 to 21 after STZ injection prevented body weight loss and improved the biochemical parameters associated with early diabetic damage and showed remarkable protective effects against renal damage. These findings indicated that ATRA protects against glomerular and tubule-epithelial hypertrophy in initial changes of DN without affecting elevation of blood glucose levels. These results are consistent with previous evidences of nephroprotective role of ATRA [13,14,15,16,17,18,35].

We next examined the mechanisms associated with fibrogenesis and the therapeutic effect of ATRA in diabetic kidney. Activated macrophages regulate fibrogenesis by providing cytokines and growth factors that modulate fibroblasts proliferation and ECM synthesis. Macrophages accumulation correlated with development of renal fibrosis in human and experimental studies of kidney diseases [26,27,36]. Consistent with these reports, we observed that in early stage of DN increased F4/80 positive cells into glomeruli and into the interstitial space. Also, we found an increase of its chemoattractant factor (MCP-1) in isolated glomeruli and isolated proximal tubules. These increments in diabetic rats were ameliorated by ATRA treatment. Therefore, our findings suggest that ATRA might prevent the progression of renal fibrosis through the regulation of macrophage infiltration in diabetic kidney. Several studies have demonstrated that ATRA might coordinate the differentiation of inflammatory macrophages (M1) to alternative macrophages (M2). M2 macrophages are involved in wound healing and immune regulation [37,38,39].

The presence of α-SMA-positive myofibroblast-like cells is considered as a hallmark for the development of kidney fibrosis [4]. In our study, we found increased localization of α-SMA-positive cells into glomeruli and into tubulointerstitial space, and its expression was elevated in isolated glomeruli and isolated proximal tubules. These results are in agreement with in vitro and in vivo studies that have shown areas of fibrosis-activated macrophages generating soluble mediators that modulate activation and proliferation of myofibroblast [27,36]. Interestingly, in our study, ATRA reduced the expression and localization of α-SMA-positive cells in glomeruli and proximal tubules. Wagner et al. [35] reported that ATRA and 13-cis-RA significantly reduced glomerular α-SMA and alleviated glomerular proliferation, glomerular lesions, and albuminuria in a Thy1.1-induced mesangioproliferative glomerulonephritis rat model I.

The progression of chronic renal insufficiency is characterized by a relentless fibrosis of the kidney affecting both the glomerular and tubulointerstitial compartments. This is characterized by an accumulation of ECM during renal fibrosis with an imbalance between the synthesis of ECM components and its degradation [5,40]. In the current study, expression of LAM β1, COL IV, and COL I was significantly higher in glomeruli and proximal tubules of diabetic rats that in nondiabetic animals. These findings suggest that overexpression of ECM molecules is not exclusive to the late stages of end-stage renal disease. As expected, treatment with ATRA reduced the levels of LAM β1, COL IV, and COL I in these segments of the nephron. Therefore, ATRA abrogates the progression of renal fibrosis by reducing the expression of ECM components. Our observations are consistent with other studies that have suggested that ATRA exerted an inhibitory effect on synthesis of pro-collagens I, III, and IV, fibronectin and laminin in liver fibrosis [41,42]. Aguilar et al. [43] reported that vitamin A deficiency can lead to the increased expression of fibronectin, laminin, and COL IV.

Several pro-sclerotic factors contribute to develop renal fibrosis; the most potent is TGF-β1, which is increased in the diabetic kidney [6,7] Consistent with these reports, we found that the expression of TGF-β1 in diabetic rats was significantly higher in glomeruli and proximal tubules than in nondiabetic groups at week 3 after STZ injection. Therefore, it is possible that TGF-β1 is a central mediator in fibrogenic processes during early DN. The in vivo data in the present study demonstrate that ATRA treatment decreased the diabetes-induced increase of TGF-β1 in isolated glomeruli and proximal tubules. This result suggests that ATRA may inhibit development and progression of renal fibrosis. Following activation of TGF-β1 signaling, the Smad effector proteins are phosphorylated and induced their nuclear translocation. Smad3 promotes renal fibrosis by directly binding to the promoter region of ECM components to trigger their production [1,7]. ATRA treatment in early DN suppressed the increased phosphorylation of Smad3 and prevented its nuclear translocation in glomerular and proximal tubular cells. In contrast, ATRA increased the expression of Smad7, an inhibitory regulator in the TGF-β1/Smad signaling pathway [6,10], which reduces the activation of TGF-β1/Smad3 signaling. These results suggest that the anti-fibrotic effect of ATRA occurs through suppression of TGF-β1/Smad3 signaling pathway in renal fibrogenesis.

Previous studies have shown that ATRA has protective effects in several experimental models of renal diseases [13,14,15,16,17,18,35]. ATRA, an active metabolite of vitamin A, exerts its effects by binding to nuclear receptor RAR, including RAR-α, RAR-β, and RAR-γ. RARs present in many tissues including kidney [13]. There is evidence that vitamin A metabolism is impaired, especially in poorly controlled diabetic patients [20,21,22]. Other reports indicate that the expression and transcriptional activation of RAR-α are significantly suppressed in cardiomyocytes treated with HG and in the hearts of Zucker diabetic rats [22,44,45]. Consistent with these studies, we showed that, under diabetic conditions, the expression of RAR-α in glomeruli and proximal tubules is decreased, suggesting that impaired RAR signaling contributes to diabetes-induced kidney injury. Most nuclear receptors are, in fact, phosphoproteins, and phosphorylation is a potent regulator of nuclear receptor function as DNA binding, dimerization, coactivator recruitment, transactivation, and degradation [13]. JNK regulates RAR-α protein levels and contributes to alterations in retinoid signaling by phosphorylation of RAR-α, resulting in ubiquitin-mediated proteasomal degradation of the receptor [29,30]. We showed that the exogenous administration of ATRA led to restoration of RAR-α expression by attenuation of its phosphorylation at serine sites and preventing interaction of RAR-α/p-JNK in glomeruli and proximal tubules. That finding indicated that the downregulated expression of RAR-α by JNK contributes to diabetes-induced fibrogenic process during early stages of DN.

## 5. Conclusions

In conclusion, ATRA treatment preserves renal function and attenuates renal hypertrophy in early-stages of renal damage in diabetic rats by inhibiting a fibrogenic process, which is not exclusive of the final stage of chronic kidney disease. This early beneficial effect occurs through (i) regulation of macrophage infiltration and localization of α-SMA positive myofibroblast-like cells into glomeruli and tubulointerstitial space, (ii) suppression of ECM molecules expression, and (iii) attenuation of the TGF-β1/Smad3 signaling in glomeruli and proximal tubules. Interestingly, diabetic condition induces activation of JNK signaling and suppression of the RAR-α expression. Therefore, ATRA/RAR-α signaling may represent a novel target for developing approaches to the treatment and prevention of renal fibrogenesis.

## Figures and Tables

**Figure 1 biomolecules-09-00525-f001:**
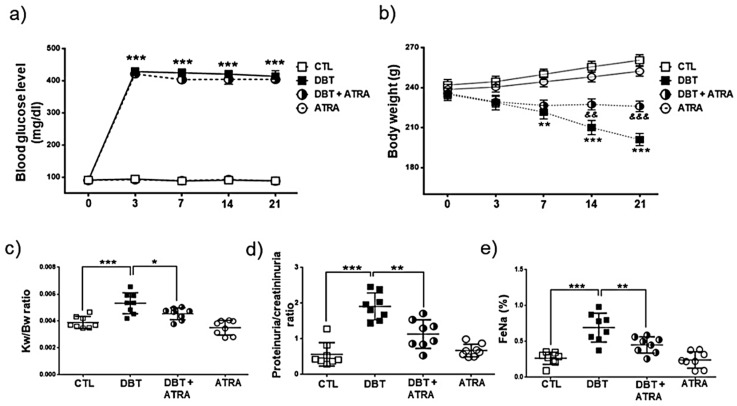
All-trans retinoic acid (ATRA) treatment ameliorated body weight loss, proteinuria, renal hypertrophy and fractional excretion of sodium induced by diabetes. (**a**) Blood glucose levels and (**b**) body weight were monitored at 0, 3, 7, 14, and 21 days after streptozotocin (STZ)-injection. At three days, rats were treated with ATRA (1 mg/kg) until day 21. ATRA did not affect hyperglycemia in diabetic groups. However, ATRA treatment prevented loss of body weight induced by diabetes. (**c**) Kidney weight/body weight (KW/BW) ratio, (**d**) proteinuria/creatininuria ratio, and (**e**) increased fractional excretion of sodium (FeNa). ATRA significantly reduced STZ-induced increase in the above parameters on day 21. Data are mean ± SD from eight rats per group. * *p* < 0.05, ** *p* < 0.01 *** *p* < 0.001 vs CTL group; && *p* < 0.01, &&& *p* < 0.001 vs DBT group.

**Figure 2 biomolecules-09-00525-f002:**
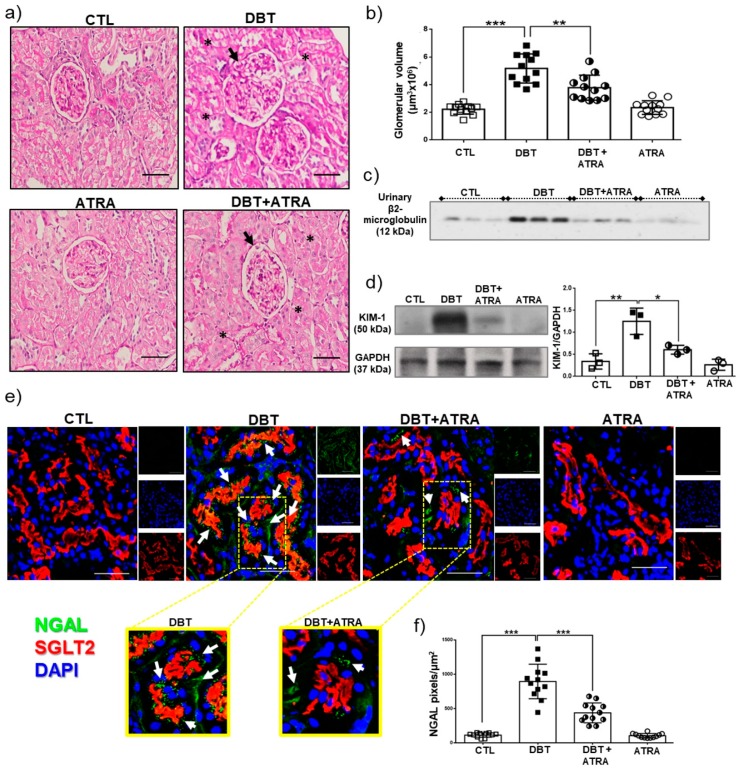
All-trans retinoic acid (ATRA) prevented increment of early markers in diabetic nephropathy. (**a**) Representative histological images of hematoxilin and eosin (H&E) staining of kidney sections from experimental animals, (**b**) glomerular volume analysis, (**c**) Western blot of urinary β2-microglobulin excretion (equal volume of urine was added from each animal), (**d**) Western blot of KIM-1 and its densitometric analysis in isolated proximal tubules, (**e**) Confocal immunofluorescence of neutrophil gelatinase-associated lipocalin (NGAL, green label), that co-localized with sodium-glucose cotransporter 2 (SGLT2), a marker of proximal tubule brush border (red label). Nuclei were counterstained with 4′,6-diamidino-2-phenylindole (DAPI, blue label), (**f**) Image-based computer assisted analysis was performed to estimate the amount of NGAL in proximal tubule sections. ATRA administration reversed morphological changes induced by streptozotocin (STZ) in rats after three weeks of hyperglycemia, including glomerular hypertrophy (black arrows), tubular lumen dilation (black asterisks), and increased expression of β2-microglobulin, KIM-1, and NGAL. Data are shown as mean ± SD (n = 12 glomeruli or proximal tubule sections per group). Glyceraldehyde 3 phosphate dehydrogenase (GAPDH) was used as loading control. Data are representative of three independent experiments, and values are expressed in mean ± SD. * *p* < 0.05, ** *p* < 0.01, *** *p* < 0.001. The scale bar is equal to 50 µm.

**Figure 3 biomolecules-09-00525-f003:**
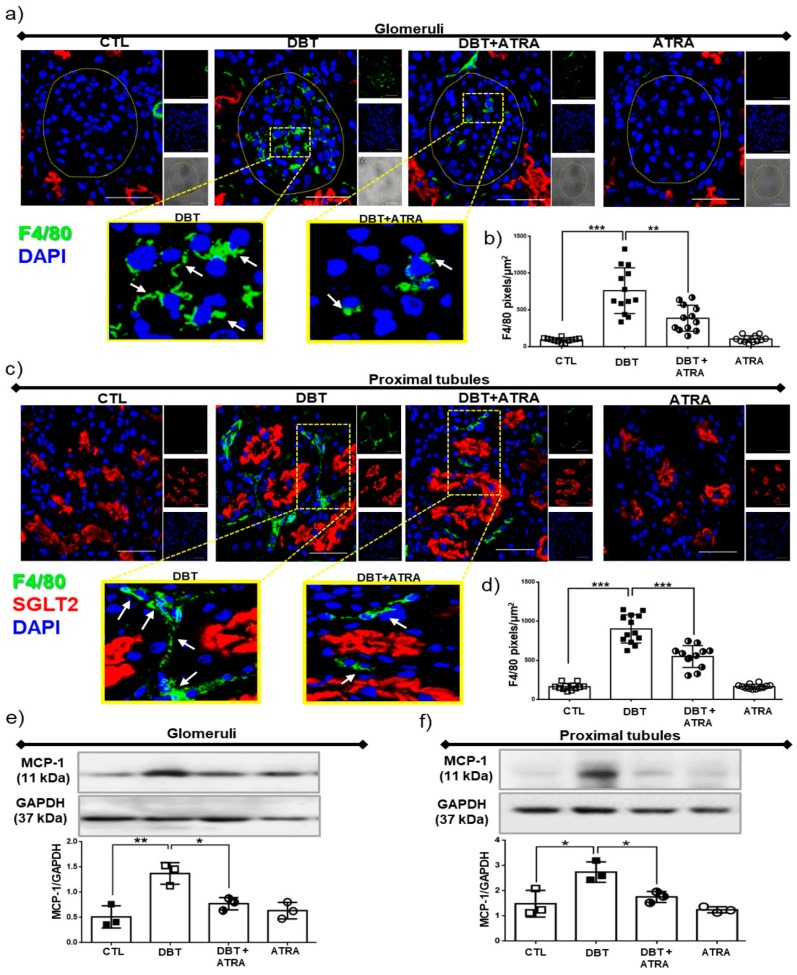
All-trans retinoic acid (ATRA) reduced the infiltration of F4/80 positive macrophages during initiation of diabetic nephropathy (DN). Confocal microscopy analysis for cellular localization of macrophages in renal sections. (**a**) Immunofluorescence images of F4/80 positive cells (green label) in glomeruli, (**b**) quantification of F4/80 positive per glomerular area (fluorescence intensity), (**c**) macrophage infiltration (F4/80 cells) into the interstitial space of proximal tubule (red label, sodium-glucose cotransporter 2, SGLT2, was used as a specific marker of proximal tubules), and (**d**) quantification of tubulointerstitial F4/80 infiltration. Nuclei were counterstained with 4′,6-diamidino-2-phenylindole (DAPI, blue label). Western blot analysis of monocyte chemoattractant protein 1 (MCP-1) in (**e**) isolated glomeruli and (**f**) isolated proximal tubules. In early stage of DN increased in renal macrophage infiltration and MCP-1 expression was elevated but was suppressed with ATRA treatment. Glyceraldehyde 3 phosphate dehydrogenase (GAPDH) was used as loading control. Data are representative of three independent experiments, and values are expressed in mean ± SD. * *p* < 0.05, ** *p* < 0.01, *** *p* < 0.001. The scale bar is equal to 50 µm.

**Figure 4 biomolecules-09-00525-f004:**
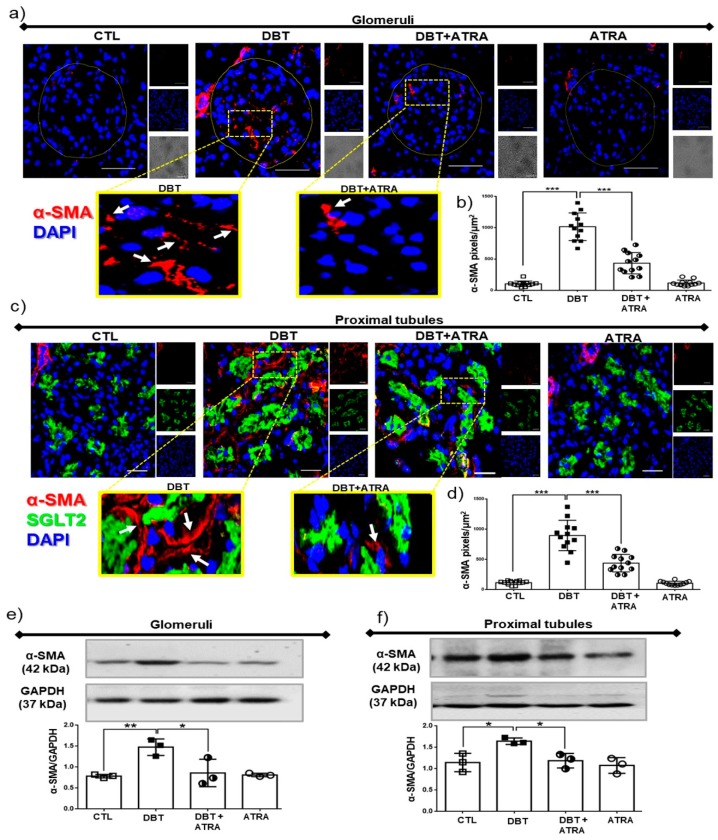
All-trans retinoic acid (ATRA) treatment decreased fibrogenic factors in early-stage diabetic nephropathy (DN). Confocal immunofluorescence for alpha-smooth muscle actin (α-SMA, marker of myofibroblasts) in renal tissue. (**a**) Representative microscopy images of α-SMA positive cells (red label) into glomeruli, (**b**) quantification of α-SMA per glomerular area, (**c**) α-SMA expression into tubulointerstitial space (red label, sodium-glucose cotransporter 2, SGLT2 was used as a specific marker of proximal tubules), and (**d**) quantification of tubulointerstitial α-SMA by fluorescence intensity. Protein expression by Western blot and densitometric analysis of α-SMA in (**e**) isolated glomeruli and (**f**) proximal tubules. Localization of myofibroblasts in kidney increased at three weeks of diabetes, and it was ameliorated with ATRA treatment. Glyceraldehyde 3 phosphate dehydrogenase (GAPDH) was used as loading control. Data are representative of three independent experiments, and values are expressed in mean ± SD. * *p* < 0.05, ** *p* < 0.01, *** *p* < 0.001. The scale bar is equal to 50 µm.

**Figure 5 biomolecules-09-00525-f005:**
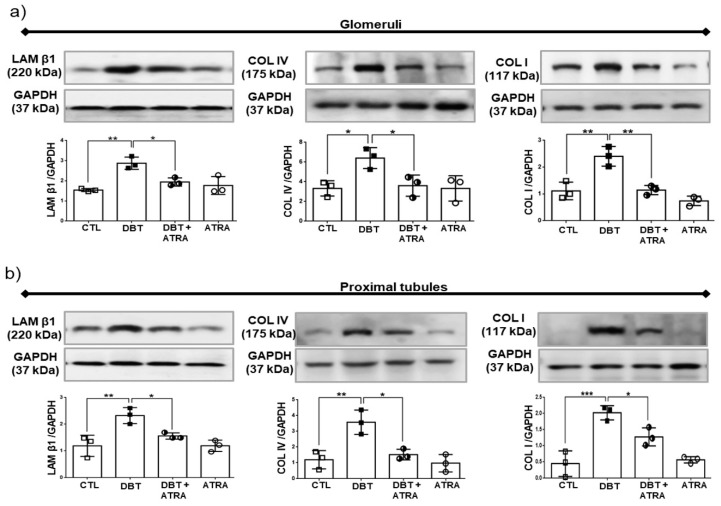
All-trans retinoic acid (ATRA) attenuated renal fibrogenesis in a rat model of diabetic nephropathy (DN). Western blot and densitometric analysis of Laminin beta 1 (LAM β1), Collagen IV (COL IV) and Collagen I (COL I) in (**a**) isolated glomeruli and (**b**) isolated proximal tubules. Laminin β1, COL I and COL IV were markedly increased during early DN. ATRA treatment decreased these fibrotic molecules. Glyceraldehyde 3 phosphate dehydrogenase (GAPDH) was used as loading control. Data are representative of three independent experiments, and values are expressed in mean ± SD. * *p* < 0.05, ** *p* < 0.01, *** *p* < 0.001.

**Figure 6 biomolecules-09-00525-f006:**
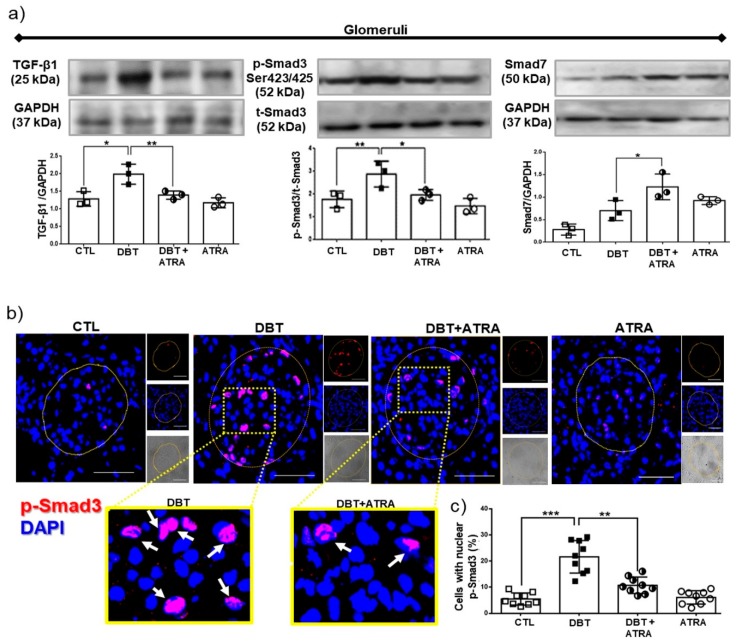
All-trans retinoic acid (ATRA) blocked activation of transforming growth factor (TGF-β1)/Similar mothers against dpp3 (Smad3) signaling pathway in glomeruli of diabetic rats. (**a**) Western blot analysis of TGF-β1, Smad3 phosphorylated (pSmad3) and Smad7 in isolated glomeruli of the four experimental groups, (**b**) confocal microscopy analysis for cellular localization of pSmad3 (red label) in renal sections. Nuclei were counterstained with 4′,6-diamidino-2-phenylindole (DAPI, blue label), and (**c**) quantification of pSmad3 nuclear translocation. ATRA treatment downregulated diabetes-induced TGF-β1 and Smad3 expression by increased Smad7. Furthermore, ATRA inhibited the nuclear translocation of pSmad3 in glomerular cells. Results are expressed as the percentage of the pSmad3 nuclei-positively stained cells into glomeruli. Glyceraldehyde 3 phosphate dehydrogenase (GAPDH) was used as loading control. Data are representative of three independent experiments, and values are expressed in mean ± SD. * *p* < 0.05, ** *p* < 0.01, *** *p* < 0.001. The scale bar is equal to 50 µm.

**Figure 7 biomolecules-09-00525-f007:**
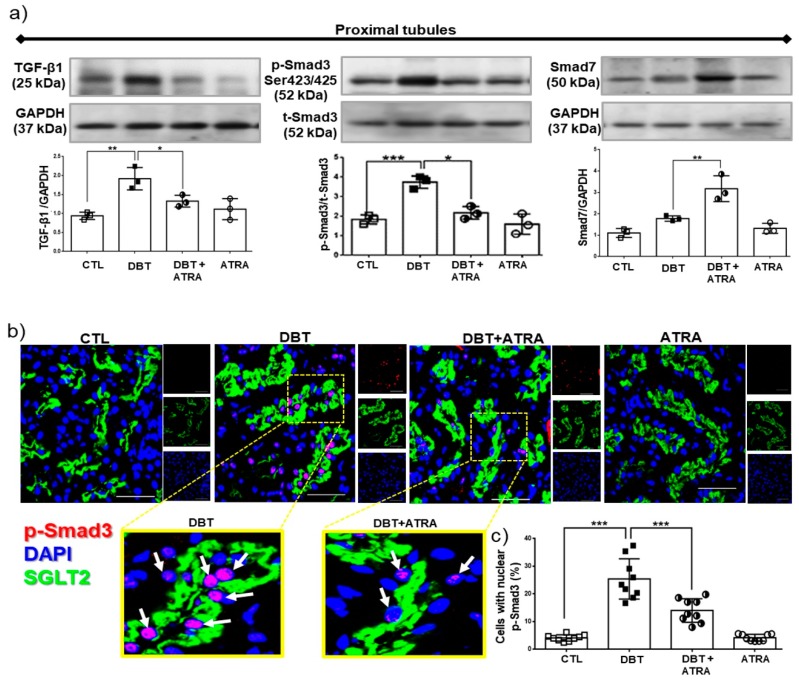
All-trans retinoic acid (ATRA) treatment attenuated the fibrogenesis processes mediated by TGF-β1/ Smad3 signaling in proximal tubules of diabetic rats. (**a**) Western blot analysis of TGF-β1, pSmad3 and Smad7 in isolated proximal tubules, (**b**) immunofluorescence analysis of pSmad3 (red label) in kidney tissue. Nuclei were counterstained with 4′,6-diamidino-2-phenylindole (DAPI, blue label), and (**c**) quantification of pSmad3 nuclear translocation. ATRA decreased TGF-β1 expression and phosphorylation of Smad3, while ATRA enhanced the expression of Smad7 in isolated proximal tubules and inhibited the nuclear translocation of pSmad3 (red label) into proximal tubules cells (green label, sodium-glucose cotransporter 2, SGLT2, was used as a marker of proximal tubule brush border). Results are expressed as the percentage of the pSmad3 nuclei-positively stained cells into glomeruli. Glyceraldehyde 3 phosphate dehydrogenase (GAPDH) was used as loading control. Data are representative of three independent experiments, and values are expressed in mean ± SD. * *p* < 0.05, ** *p* < 0.01, *** *p* < 0.001. The scale bar is equal to 50 µm.

**Figure 8 biomolecules-09-00525-f008:**
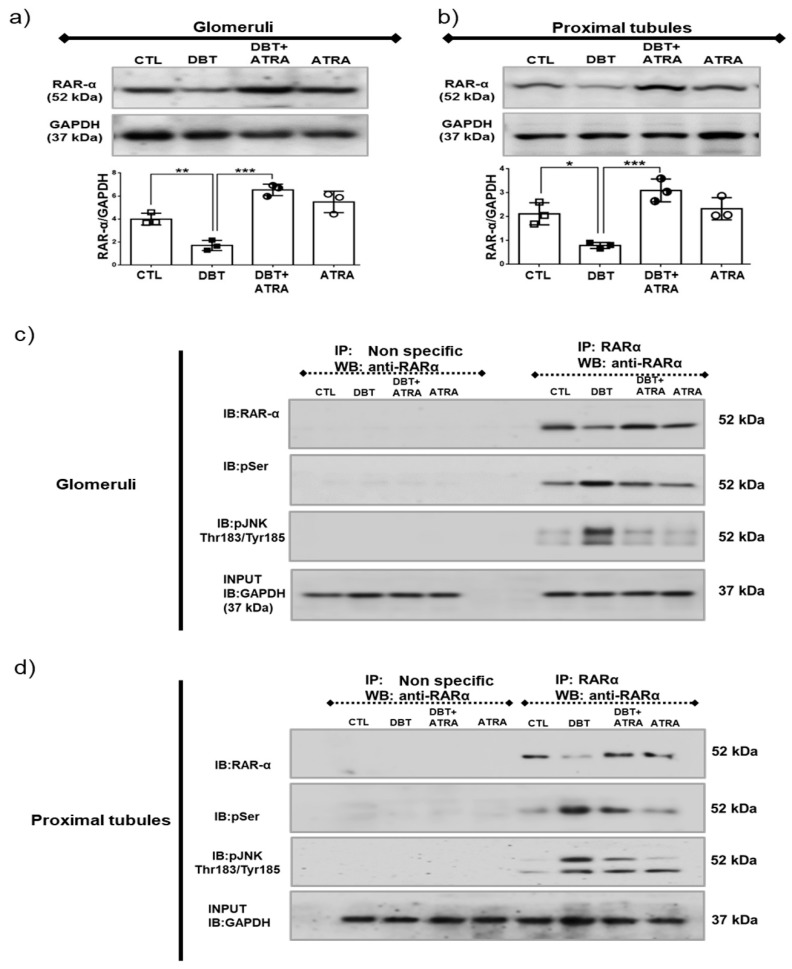
All-trans retinoic acid (ATRA) inhibited STZ-induced phosphorylation and loss of retinoic acid receptor alpha (RAR-α) expression by c-Jun N-terminal kinase (JNK). Western blot analysis of RAR-α in (**a**) in isolated glomeruli and (**b**) proximal tubules. To evaluate whether hyperglycemia-induced degradation of RAR-α is regulated by its phosphorylation, cell lysates of (**c**) isolated glomeruli and (**d**) isolated tubules were studied by immunoprecipitation (IP) with antibody to RAR-α, followed by immunoblot (IB) analysis with anti RAR- α and anti-phosphoserine antibody. To evaluate that JNK promotes phosphorylation of RAR-α IP analyzes were performed. Diabetes downregulates the expression of RAR-α by its phosphorylation in serine residues mediated activation of JNK. ATRA treatment decreased these changes. As shown in panels (**c**) and (**d**), no signal was found under nonspecific conditions of IP performed with an unrelated antibody. GAPDH was evaluated in input extracts as loading control. Data are representative of three independent experiments, and values are expressed in mean ± SD. * *p* < 0.05, ** *p* < 0.01, *** *p* < 0.001.

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
