# Peer review of "All-Trans Retinoic Acid Attenuates Fibrotic Processes by Downregulating TGF-β1/Smad3 in Early Diabetic Nephropathy"

_biomolecules, 2019, doi:10.3390/biom9100525_

Round 1
Reviewer 1 Report
The authors have described about the effect of ATRA on diabetic nephropathy. The observation is interesting and important. Although ATRA activates RAR, its metabolite 9-cisRA activates both RAR and RXR. Therefore, the authors need to perform same experiments using 9-cisRA, at least for several key data.
Author Response
"Consulte el archivo adjunto."
Reviewer 2 Report
Reviewer Comments
This manuscript discussed the role of All-trans retinoic acid in preventing fibrosis through TGF-β1/Smad3 signaling in a diabetic nephropathy model. The authors have done extensive study to provide adequate conclusions but there are some missing links that must be addressed. So, the authors should consider the following comments for major improvement prior to re-submission.
In the manuscript, introduction needs to be improved. Methods in 2.7, 2.8, and 2.12 needs to be more descriptive. Kim-1 as marker of renal injury is not shown. Include in the manuscript to substantiate the data. 2a Indicate with arrows the site of injury or fibrotic areas and the areas which get better after ATRA treatment. 2c. What assay is this? Is it a western blot? If yes, then it has to be normalized to actin, gapdh, or total protein and properly quantified. 2d. Images are not clear and representative. Please provide better images and indicate with arrows wherever necessary. 6a. the western blot showing TGF-β1 is not indicative of what shown in quantification. There seems to be no decrease in the above molecule after ATRA treatment in diabetic rats. Also, the p-Smad3 western blot should be quantified with total Smad. Include total Smad. 8c and 8d. The western blot should be replaced. There are total 8 samples. Where from the extra bands are coming? Repeat the IP and provide a proper representative image. The qPCR data must be shown for important molecules in TGF-β1/Smad3 pathway. What happens to Smad-2? Indicate the pathologies observed during early stage of diabetic nephropathy and score it quantitatively. How ATRA treatment improved the condition? Include in your discussion. Also, show other markers of Glomerular injury especially diabetic nephropathy such as WT-1, Synaptopodin etc. Refer Modulation of apolipoprotein L1-microRNA-193a axis prevents podocyte dedifferentiation in high-glucose milieu (American Journal of Physiology-Renal Physiology) https://doi.org/10.1152/ajprenal.00541.2017. Schematic graphic showing the overall theme of the manuscript is missing. Discussion needs to be improved with most recent references. All the statistical analysis should be checked thoroughly. The manuscript should be thoroughly checked for grammatical and typing errors.

Author Response
"Consulte el archivo adjunto."

Round 2
Reviewer 1 Report
I still think it is important to perform key experiments using 9-cis RA (or receptor specific retinoids).